# An Overview of Diagnostic and Management Strategies for Talaromycosis, an Underrated Disease

**DOI:** 10.3390/jof9060647

**Published:** 2023-06-06

**Authors:** Silvere D. Zaongo, Fazhen Zhang, Yaokai Chen

**Affiliations:** 1Department of Infectious Diseases, Chongqing Public Health Medical Center, Chongqing 400036, China; zsildieu@yahoo.fr; 2Fifth Unit for Tuberculosis, Chongqing Public Health Medical Center, Chongqing 400036, China; zhangfazhen@163.com

**Keywords:** *T. marneffei*, talaromycosis, diagnosis, treatment, prognosis

## Abstract

Underrated and neglected, talaromycosis is a life-threatening fungal disease endemic to the tropical and subtropical regions of Asia. In China, it has been reported that talaromycosis mortality doubles from 24 to 50% when the diagnosis is delayed, and reaches 100% when the diagnosis is missed. Thus, the accurate diagnosis of talaromycosis is of utmost importance. Herein, in the first part of this article, we provide an extensive review of the diagnostic tools used thus far by physicians in the management of cases of talaromycosis. The challenges encountered and the perspectives which may aid in the discovery of more accurate and reliable diagnostic approaches are also discussed. In the second part of this review, we discuss the drugs used to prevent and treat *T. marneffei* infection. Alternative therapeutic options and potential drug resistance reported in the contemporary literature are also discussed. We aim to guide researchers towards the discovery of novel approaches to prevent, diagnose, and treat talaromycosis, and therefore improve the prognosis for those afflicted by this important disease.

## 1. Introduction

Even in this present era of the COVID-19 pandemic, it remains necessary to be vigilant to neglected infections that are and have been present even before the onset of the pandemic. One of these infectious diseases, which remains underrated today, is talaromycosis. Talaromycosis (formerly known as Penicilliosis [1]) is an infection caused by *Talaromyces marneffei,* a thermally dimorphic fungus formerly known as *Penicillium marneffei*. In 1956, the first case of *T. marneffei* infection in bamboo rats was studied [2,3], and subsequently in 1973, the first case in humans was reported [4]. Talaromycosis is an invasive fungal infection which can be localized to the upper or lower respiratory tract [5,6,7], bones, joints, and intestinal tract, or disseminated across multiple organ systems [8]. The main clinical manifestation of talaromycosis is skin lesions which are characterized by raised bumps (usually small and painless) [9,10] on the skin, particularly on the face, the neck, and the extremities [11,12]. Notably, HIV-negative individuals with talaromycosis are less likely to have skin lesions [9,13]. In the particular context of advanced HIV infection (patients with CD4+ T-cells < 200 cells/µL), talaromycosis disseminates to organs such as the lung, liver, spleen, gastrointestinal tract, blood stream, and bone marrow [8,13,14]. Interestingly, some reports have indicated that *T. marneffei* may cause primary pulmonary talaromycosis even in apparently healthy individuals [15,16], which suggests that talaromycosis may well be a more common cause of pneumonia in endemic areas than has been hitherto assumed. This information warrants close investigation even if pneumonia is considered to be a rare illness among *T. marneffei*-infected patients, as such cases of respiratory tract talaromycosis have, indeed, been reported in the literature [17,18].

Talaromycosis is a neglected tropical disease that is endemic to the tropical and subtropical regions of Asia. It mainly affects individuals with HIV or other immunosuppressive conditions (cancer, organ transplant recipients, and auto-immune diseases) [19]. In some rare cases, healthy individuals can also develop talaromycosis [15]. It is believed that the disease is spread to humans via the inhalation of *T. marneffei* spores existing in the environment [11], although robust evidence supporting such a transmission method has not, as yet, been revealed. Hyperendemic areas are localized in Southeast Asia (Thailand, Vietnam, and Myanmar), East Asia (southern mainland China, Hong Kong, and Taiwan), and Northeastern India [11] and have experienced a rapid rise in incidence due to the ongoing HIV pandemic, which remains a significant public health burden in these regions. In addition to HIV, which depletes the immune system, the increasing use of biological medicines such as corticosteroids, immune checkpoint inhibitors, and tumor necrosis factor inhibitors also increases the risk of invasive fungal infections [20,21] including talaromycosis. Thus, talaromycosis is the third most common opportunistic infection (OI) in countries such as China, Thailand, and Vietnam [13,14]. Moreover, it has been shown that talaromycosis is inextricably linked to poverty, and the endemic regions tend to invariably be low-income and lower-middle-income countries [22]. Studies have demonstrated that talaromycosis preferentially affects agricultural workers in rural areas, suggesting that a history of occupational or other long-term exposure to soil may be associated with disseminated *T. marneffei* infections [23]. According to the findings of some researchers, compared to non-farmers, farmers have a 70–90% greater risk of developing talaromycosis [23,24]. Additionally, residents in highland communities have been shown to have a three-fold greater risk of disease compared to non-highland communities [24]. Although talaromycosis is endemic to the preceding areas and settings, its threat is not limited to specific countries or geographical regions, particularly in this era where the number of international travelers and instances of large population movements are increasing. It is known that travelers who visit areas of endemicity are also potentially vulnerable [8,25,26,27,28]. Furthermore, talaromycosis tends to predominantly affect young people during the peak years of their economic productivity [8,13,14]. By mid-2022, over 288,000 cases of talaromycosis had been reported in 34 countries [29]. Interestingly, the prevalence of talaromycosis in the general population is unknown, but the pooled prevalence in people living with HIV (PLWH) is estimated at 3.6%, with a range of between 0.1% and 19.6%, depending on specific regions and countries [30]. Most importantly, it is generally acknowledged that talaromycosis is a disease with a high case–fatality rate which kills up to a third of diagnosed individuals [13,14], and is a leading cause of HIV-associated death, surpassing mortality attributable to tuberculosis and cryptococcal meningitis in this population [13,14].

In light of the preceding information, it is imperative to, therefore, consider *T. marneffei* infection as a major public health issue, which preferentially targets individuals (i) with immunosuppressive conditions (especially HIV infection), (ii) living in poor socioeconomic settings, and (iii) within their peak years of economic productivity (particularly young farmers). Although *T. marneffei* infection has been ranked the “third most feared” fungal infection in 2018 by Hyde et al., in their review of the ten most feared fungal pathogens globally [31], the diagnosis and treatment of talaromycosis has, as yet, not received adequate or meaningful global attention thus far. Indeed, basic information, such as the prevalence in the general population remains unknown [30]; however, mortality remains stable in endemic regions, such as China [14]. Alarmingly, Hu et al. [32] have demonstrated that talaromycosis mortality (i) doubles from 24 to 50% when the diagnosis is delayed and (ii) approaches 100% when the diagnosis is missed. In the particular case of HIV-positive individuals, it has been demonstrated that early ART initiation helps to reduce the mortality due to talaromycosis (2.2% in early ART group vs. 8.9% in deferred ART group) [33]. A study by Zhou et al. also found that talaromycosis is one of the predictors for death due to AIDS in Chongqing, China [34]. We, therefore, propose extensively reviewing the diagnosis and treatment of talaromycosis, which are the patently challenging aspects of *T. marneffei* infection, in order to propose workable solutions to control the spread of the disease, and thus limit its burden on affected populations.

## 2. Diagnosis of Talaromycosis

### 2.1. Current Approach—Culture-Based Diagnosis

The current approach for diagnosis is mainly represented by blood culture, which takes up to 14 days for diagnostic identification. In addition to this huge diagnostic time gap, blood culture can also miss up to 50% of infections, and only detects talaromycosis in its advanced stage, when *T. marneffei* becomes widely disseminated [13,14,35]. In general, the culture-based diagnostic approach tends obviously to have high specificity, high accuracy, and wide applicability to a variety of clinical specimens (skin, bone marrow, or lymph node biopsies can also be used). Depending on the specimen, the diagnostic accuracy for laboratory culture may vary. Data suggest an accuracy of 34% and 90% for sputum and skin biopsy, respectively [10]. Furthermore, an accuracy as high as 100% has been reported when the culture has been performed using bone marrow or lymph node biopsy [10]. However, culture-based diagnosis negatively affects the clinician’s decision for the management of therapy. Indeed, culture-based diagnosis, which is considered the gold standard for the diagnosis of *T. marneffei,* requires up to two to four weeks for detectable growth to occur [36]. Furthermore, it is worth noting that *T. marneffei* culture is temperature and medium dependent. As such, it has been shown that the optimal temperature for growth of the mycelial form is 25 °C, while the yeast form grows well at 37 °C [37]. The yeast form is cultured on brain heart infusion (BHI) culture medium and the mycelial form grows on Sabouraud dextrose agar, where it has a greenish yellow color associated with a red diffusible pigment [38]. It is, therefore, imperative to explore other viable rapid methods of diagnosis which do not unduly delay the initiation of appropriate antifungal therapy in patients with early active talaromycosis infection who require and would benefit from rapid diagnosis and treatment.

### 2.2. Microscopic Analysis

Other than culture-based diagnosis, a presumptive diagnosis of talaromycosis may be made on the basis of microscopy. This approach may provide results more rapidly than culture-based diagnosis. Thus, laboratory staining of pathological specimens may be performed, and histological studies may be carried out on these specimens. For example, Grocott’s methenamine silver or peripheral acid–Schiff staining of histopathological sections will reveal *T. marneffei* yeast cells (round or oval) present within macrophages. When specimens are obtained via (i) fine-needle aspiration of lymph nodes or bone marrow, (ii) touch smears of skin, or (iii) lymph node biopsy, Wright’s staining will reveal clear basophilic, spherical, oval and elliptical yeast cells possessing a central septation (in contrast to budding, which is generally utilized by other clinically significant yeasts), which is quite specific to *T. marneffei* [11,39,40]. Nevertheless, microscopic analysis possesses several disadvantages. For example, a skin smear can be analyzed microscopically to detect the presence of *T. marneffei*; however, skin lesions are absent in 30–60% of patients with talaromycosis [35,41], which further complicates the identification of *T. marneffei* on skin. Additionally, it is worth mentioning that microscopic analysis requires well-trained professional laboratory personnel, with at least two independent analysts required to validate the observed results.

### 2.3. Serological Antigen or Antibody Detection

The immunohistochemical detection of the monoclonal antibody EB-A1, which is a monoclonal antibody to galactomannan (GM found in *Aspergillus* spp. and *T. marneffei*, can also be used as a diagnostic tool [42,43]. It is known that the GM assay has a cross-reactivity for other fungal species such as *Cryptococcus neoformans*; however, a study by Huang et al. [43] has observed that a significantly higher GM assay optical density index is seen when *T. marneffei* is involved. This can be considered to be a useful tool for early diagnosis of talaromycosis, particularly in regions where talaromycosis is endemic. Additionally, an elevated serum level of beta (β)-D-glucan has been observed in 82% of talaromycosis cases in Japan [44]. In the preceding study, the patients travelled to countries where talaromycosis is endemic. Classically, the β-D-glucan assay is a serodiagnostic method for invasive fungal infections such as aspergillosis and candidiasis [45]; however, if the profiling of β-D-glucan can potentially detect 82% of talaromycosis cases, it is likely that the β-D-glucan assay may be a viable candidate for the clinical diagnosis of *T. marneffei*-infected patients in non-endemic regions who have a history of travel to endemic regions. It is worth mentioning, however, that the cohort size of the preceding Japanese study was very small (11 patients), and as such, this study may not have been adequately powered to generate conclusions with a reliably high degree of power.

Two promising new approaches to talaromycosis diagnosis have been developed. These are monoclonal antibody (m-Ab)-based antigen detection enzyme immunoassays, viz., the yeast-phase-specific monoclonal antibody 4D1 (mAb-4D1) and mAb-Mp1p. Results gleaned from small studies [46,47] have shown that the mAb-4D1enzyme immunoassay and its immunochromatographic platform have high sensitivity and specificity for talaromycosis, as observed by Narayanasamy et al. [12]. On the other hand, the mAb-Mp1p assay, which has been more extensively studied compared to the mAb-4D1 assay, has shown a sensitivity of between 75 and 86% and a specificity between 98 and 99% [36,48]. Of particular interest is the fact that the mAb-Mp1p assay has been observed to be more sensitive than blood culture (85% vs. 73%, respectively) [48]. Furthermore, the diagnostic efficacy of the mAb-Mp1p assay depends on the specimen used for the analysis. The sensitivity of the mAb-Mp1p assay has been observed to be higher when using urine samples than in plasma samples. When both plasma and urine were tested, the mAb-Mp1p assay sensitivity was higher than that in urine alone [48]. In light of these promising results, the mAb-Mp1p enzyme immunoassay is currently being evaluated in one multicenter prospective study (NCT04033120) as a rapid diagnostic test for *T. marneffei* diagnosis. In China, the mAb-Mp1p enzyme immunoassay was approved for use in October 2019 [12].

### 2.4. PCR-Based Approaches and Metagenomic Next-Generation Sequencing

To provide a rapid diagnosis, PCR-based approaches have been developed for various diseases. Thus, primers have been created based on species-specific DNA sequences, such as the internally transcribed spacer of the 5.8S rRNA [49], the 18S rRNA [50], and MP1 [51] genes. Moreover, various amplification methods may be utilized, including single PCR, nested PCR, and one tube semi-nested PCR [12]. Although these molecular assays possess both high sensitivity (ranging from 10 to 100%) and high specificity (>95%) [52], they remain expensive and technically complex to perform. Nevertheless, in the particular case of talaromycosis, where a timely diagnosis is essential for a favorable patient prognosis, molecular assays have recently attracted much attention.

In contemporary times, the development and utilization of metagenomic next-generation sequencing (mNGS) has greatly contributed to the detection of pathogenic microorganisms [53,54,55]. This powerful tool can simultaneously identify bacteria, fungi, viruses, and parasites with both high efficiency and high sensitivity [56,57]. In an investigative approach exploring the performance of mNGS for the diagnosis of *T. marneffei* infection, Liu et al. [58] have observed that the sensitivity and specificity of mNGS was 100% and 98.7%, respectively. Compared to culture-based methods, mNGS did not only show advantages with respect to high detection efficiency, but was also quicker to perform. Indeed, Liu et al. have shown that mNGS required just 26 h to complete and obtain a diagnosis, while when blood culture and histopathology were used, 3–14 days and 6–11 days, respectively, were required. Previous publications [59,60] have shown that mNGS provides faster and more accurate diagnosis of *T. marneffei* infection in clinical settings. Most importantly, mNGS can be applied even on formalin-fixed and paraffin-embedded (FFPE) samples, as reported by Zhou et al. [60] when diagnosing a case of gastrointestinal *T. marneffei* infection with negative blood culture. Thus, mNGS is likely to be of significant importance for the future diagnosis of talaromycosis, as a rapid diagnosis facilitates timeous and robust management of the infection, with administration of appropriate drugs before wide dissemination of *T. marneffei*, with its life-threatening consequences, can occur.

A summary of the methods used to diagnose *T. marneffei* infection, including their advantages and their limitations, is provided in Table 1.

### 2.5. Challenges and Perspectives for the Diagnosis of Talaromycosis

It is known that patients may develop talaromycosis some years after *T. marneffei* exposure [26,62,63]. In other words, *T. marneffei* may remain latent within the host and may reactivate at a later stage. Unfortunately, there are no animal experiments and no clinical evidence to assist in explaining the mechanisms which underlie latency. In vivo options for an animal model, such as mouse, zebrafish, *Galleria mellonella*, and *C. elegans,* have all been explored without much success. Indeed, *C. elegans* is not a viable option, as it cannot withstand human physiological temperature. The *Galleria mellonella* moth can be grown at 37 °C; however, it does not possess an adaptive immune response system [64]. The zebrafish has been considered to be a relatively ‘good’ model due to its physiological and immunological resemblance to humans. Thus, pathogen-host infection platforms have been developed utilizing the zebrafish to investigate the virulence of fungal pathogens such as *A. fumigatus* [65], *C. albicans* [66], and *C. neoformans* [67]. Recently, researchers have focused their interest on using a zebrafish model of talaromycosis to demonstrate the function of innate immune cells during *T. marneffei* infection. Compared to the preceding model options, mice are considered the most appropriate model for mimicking the natural infection cycle in humans. However, several unknown and poorly understood factors, such as the required pathogenic dose, the method and route of infection, and the underlying immunological condition may possibly distort the outcomes of these investigations in mice. As there is currently no standard animal model accepted by all, the development and adoption of a suitable infection model is critical. In a similar manner, the development of tools oriented towards the prediction and/or rapid detection of *T. marneffei* infection based on the host response are urgently required.

*T. marneffei* is considered to be a facultative intracellular pathogen. Thus, it is found within macrophages and tissues of infected patients [11,68]. Therefore, ex vivo macrophage cell lines, for example, may be used to study the immunopathogenesis of *T. marneffei.* In considering that *T. marneffei* uses its ability to hide within macrophages to evade the host immune system for years, an approach using single cell sequencing (scRNA-seq) may help to predict and identify particular signatures (micro-RNA, proteins, receptors, cytokines, etc.) expressed by the infected cells. The best approach may well be to regularly collect macrophages from blood samples of HIV-positive and HIV-negative individuals for a relatively sustained period of time. Then, for those who develop *T. marneffei* infection during the period of collection, a comparative analysis of the macrophage proteome and genome signatures would assist in identifying potential useful biological markers that indicate *T. marneffei* infection. For example, it has been reported that patients with *T. marneffei* infection display cytokine disorders [69]. Therefore, Wang et al. [29] have suggested that the level of serum anti-interferon-gamma (IFN-γ) may be a simple and rapid method to diagnose *T. marneffei* infection in HIV-negative patients. However, further investigations into this method are warranted, as it is known that coinfections of *T. marneffei* with nontuberculous mycobacteria, *Mycobacterium tuberculosis*, and *C. neoformans* exist in both HIV-infected and HIV-negative individuals. These pathogens share common clinical manifestations (e.g., increased anti-IFN-γ antibody levels in the host), which unfortunately makes the preceding approach to identify *T. marneffei* infection much more challenging. Nevertheless, researchers have noted that in Southeast Asia, HIV-negative patients who possess the HLA-DR*15:02/16:02 and HLA-DQ*05:01/05:02 genes are predisposed to *T. marneffei* infection [29]. In HIV-infected individuals, a prediction model has been developed to produce accurate clinical outcome assessments based on the associations between *T. marneffei* infection and the clinical features of HIV infection [70]. Thus, candidate factors such as skin lesions, AST level, ALT ratio index, peripheral or abdominal lymphadenopathy, and CD4+ T-cell counts were all considered in different prediction models. It was observed that the model combining AST levels and the ALT ratio index displayed high power, specifically for the early diagnosis of *T. marneffei* infection [70]. Although this work has only been published in a preprint research platform, the approach used to conduct the investigation is impressive, and may well be improved upon using a higher volume of, and more frequently collected data obtained from clinical practice. Furthermore, Qin et al. [71] have found that age, AST/ALT ratio, albumin levels and blood urea nitrogen levels can predict the risk of death in HIV-infected patients with talaromycosis. These variables could, therefore, be seen as predictors of talaromycosis upon further investigations. Lastly, it is significantly important to note that genotype TLR2 rs1339 and rs7656411 polymorphisms in patients living with HIV are more frequently associated with talaromycosis [29]. This information may be important for the development of screening approaches oriented towards HIV-positive individuals.

Talaromycosis is curable if the clinical diagnosis is accurate and is made timeously in the early stages of the infection. In order to facilitate this, a profound comprehension of the intricate interplay between *T. marneffei* with the host immune system is essential, as this is likely to assist in developing new diagnostic approaches. Despite the lack of workable animal models at present, the information gleaned from studies with macrophages (Figure 1) may help to identify potential biomarkers. In addition to macrophages, neutrophils are known to partially use the myeloperoxidase enzyme to eliminate fungal infections [72,73], and may provide an interesting approach for the diagnosis of *T. marneffei* infection. Indeed, determining the levels of myeloperoxidase in people susceptible to talaromycosis (HIV-positive, living in endemic areas, and young people) may serve as one diagnostic parameter. It is likely that a deficiency in myeloperoxidase levels indicates a proliferation of *T. marneffei,* as is seen with *C. albicans* [72].

To determine the presence of *T. marneffei* infection, techniques using antifungal susceptibility testing can also be utilized. Thus far, there is no standard method for this approach to diagnosis, and antifungal drug susceptibility may differ between mycelial and yeast forms [76,77]. In 2018, Lei et al. [78] observed that the YeastOne^TM^ YO10 assay may be a viable option for susceptibility testing, particularly for the susceptibility profile of echinocandins, azoles, and amphotericin B against the yeast phase of *T. marneffei*. A recent publication by Fang et al. [79] confirmed the observations of Lei et al. [78], which observed that the yeast forms of the *T. marneffei* isolates are inhibited by amphotericin B and azoles (itraconazole and voriconazole with MIC_50_ and MIC_90_ ≤ 0.015 μg/mL). Furthermore, Fang et al. [79] have shown the antifungal activity of triazoles to be superior to echinocandins and 5-fluorocytosine for the yeast phase of *T. marneffei*. This information may be useful to determine the presence of *T. marneffei*, and the evolution of the strains in terms of resistance to antifungal drugs. In addition to the YeastOne^TM^ YO10 assay, Fang et al. [79] proposed using matrix-assisted laser desorption/ionization time-of-flight mass spectrometry (MALDI-TOF MS) as a tool to identify and cluster *T. marneffei* isolates.

Diagnostic timing is paramount in talaromycosis. An early diagnosis is essential for the treatment strategy to be successful. In the following section, we present the therapeutic approaches used to treat talaromycosis, as reported in the contemporary literature.

## 3. Treatment

To date, most therapeutic strategies reported in the literature focus on HIV-infected individuals. Moreover, there are no standardized recommendations regarding the appropriate duration of treatment and prophylaxis for *T. marneffei* infection among the HIV-uninfected. However, insights from the literature reveal that the treatment duration may be significantly longer in HIV-uninfected individuals compared to HIV-infected patients. In HIV-infected individuals, treatment for talaromycosis should be followed by prophylactic treatment to prevent relapse. Currently, as recommended by Tun et al. [80], secondary prophylaxis can be safely stopped in patients with talaromycosis after immune reconstitution, with a sustained increase in CD4+ T-cell counts to ≥ 100 cells/µL after ART initiation. In some cases, it is seen that talaromycosis requires lifelong treatment in HIV-uninfected individuals [8]. In the following discussion, therapeutic strategies used to prevent and treat talaromycosis are reviewed.

### 3.1. Prophylactic Approaches

It is known that prophylactic measures help to reduce the incidence of talaromycosis. For example, one controlled trial has observed that itraconazole (200 mg per day) can be an efficacious prophylactic drug against talaromycosis and other invasive fungal infections in HIV-1 infected individuals [81]. However, itraconazole prophylaxis was found not to be associated with a survival advantage when given to patients with advanced HIV disease. The preceding evidence was reported 20 years ago; however, itraconazole has not been widely used in clinical practice due to concerns raised with respect to its toxicity, drug–drug interactions, and cost [12]. Nevertheless, primary prophylactic treatment with itraconazole (200 mg per day until CD4+ T-cell counts exceed 100 cells/µL for more than 6 months) is recommended for HIV-infected individuals with CD4+ T-cells counts < 100 cells/µL living in or traveling within endemic areas [82]. Other prophylactic options exist, which can also be regarded as potential therapeutic options. Indeed, in vitro studies [2,83,84,85] using different investigative approaches have shown that posaconazole, voriconazole, itraconazole, and other azole drugs have high activity against *T. marneffei*, while amphotericin B and echinocandins show intermediate and low antifungal for *T. marneffei* activity, respectively. Further studies need to be conducted to evaluate the efficacy of these drugs (other than echinocandins, which do not show promising results) in preventing *T. marneffei* infection.

A prophylactic strategy similar to that used for cryptococcosis [86] is currently considered for *T. marneffei* infection. This approach is based on the administration of co-trimoxazole, a microbial agent which targets microbial folate biosynthesis [87,88], as a prophylactic drug. This has now become an option, as one retrospective study by Jiang et al. [89] observed that patients on co-trimoxazole prophylaxis (a total of 3359 persons receiving ART from 2005 to 2016) had a significantly lower *T. marneffei* infection rate than those not on co-trimoxazole (4.11% versus 7.53%, respectively). Furthermore, co-trimoxazole was observed to inhibit *T. marneffei* growth in an ex vivo THP1 macrophage model [90]. The reported mechanism indicated that co-trimoxazole blocks dihydropteroic acid synthase (DHPS), dihydrofolate synthase (DHFS), and dihydrofolate reductase (DHFR), which are all utilized by *T. marneffei* to promote growth. Additional investigations (epidemiological studies in particular) are required to validate the potential use of co-trimoxazole as a safe and effective prophylactic drug against talaromycosis.

The assessment of a prophylactic approach requires an efficient diagnostic approach. However, even now, the diagnosis of talaromycosis remains challenging. For example, it has been demonstrated that in cryptococcosis, a diagnosis based on antigen screening combined with fluconazole administered as prophylactic agent prevents cryptococcal meningitis, reduces mortality [91], is cost-effective [92], and is largely adopted for treatment of HIV-infected individuals across the world [93,94]. This approach could be of particular use for the control of talaromycosis as *T. marneffei* antigenemia, which is associated with mortality within 12 months [95], has been shown to precede the development of culture-confirmed talaromycosis by up to 16 weeks [96,97]. Further investigations delving into this specific area of research interest are necessary and will provide a more accurate assessment of prophylactic drug choice.

### 3.2. Curative Approaches

Amphotericin B deoxycholate (D-AmB) remains the first-line antifungal treatment for severe *T. marneffei* infection. In general, the treatment consists of amphotericin B administration [98] followed by weeks to months of azoles such as itraconazole, voriconazole, and posaconazole [99]. In the particular case of HIV-infected individuals, the international guidelines recommend the administration of D-AmB at 0.6–1.0 mg/kg per day for 2 weeks. As D-AmB is not always well tolerated, liposomal amphotericin B (L-AmB, 3–5 mg/kg per day for 2 weeks) is an effective and better tolerated alternative [100]. For AIDS patients, it is recommended to prescribe intravenous amphotericin B at 0.6 to 0.7 mg/kg of body weight or 3 to 5 mg/kg of L-AmB daily for 2 weeks [98].

Other than amphotericin B, itraconazole (400 mg per day for 10 weeks, then 200 mg per day for at least 6 months until CD4+ T-cell counts exceed 100 cells/µL) can be administered [101,102]. In 2017, results from one clinical study revealed that D-AmB therapy was superior to itraconazole (at 600 mg per day for 3 days, followed by 400 mg per day, for 11 days) as initial treatment for talaromycosis [103]. Indeed, the risk of mortality at week 24 was 11.3% in the D-AmB group and was 21.0% in the itraconazole group (*p* = 0.006), and treatment with D-AmB was associated with significantly more rapid clinical resolution and fungal clearance. In cases where patients do not respond to amphotericin B initial therapy, voriconazole can be administered (6 mg/kg BID for one day, 4 mg/kg BID for 10–14 days, followed by oral voriconazole 200 mg BID for 12 weeks), particularly in cases of disseminated talaromycosis [35,104]. In parallel, Zhou et al., have demonstrated that therapy using D-AmB and voriconazole had similar efficacy on HIV-infected individuals with talaromycosis [105]. However, in this prospective multicenter cohort study, it is worth noting that compared to voriconazole, D-AmB led to significantly higher clinical resolution in ART-naïve patients [105].

Furthermore, in some studies, 70–80% of HIV-uninfected children with talaromycosis have been successfully treated with voriconazole (7 mg/kg twice per day IV for at least 12 days, followed by oral voriconazole for at least 13 weeks), and not a single adverse event was recorded during or after the treatment course [104,106]. In one report, it has also been shown that D-AmB 1 mg/kg/d can be used to treat *T. marneffei*-infected children. However, D-AmB presented significant nephrotoxic side effects [107]. Talaromycosis in children in general is quite a specific public health issue, as in adults either with or without HIV-infection, the case fatality rate is estimated at 20% and 29%, respectively, whereas mortality in pediatric patients may be up to 50% [108].

### 3.3. Challenges Encountered in the Treatment of Talaromycosis

#### 3.3.1. The Need for Alternative Options

Althoug, it is known that amphotericin B is excellent for the treatment of talaromycosis, its high cost and difficulties in gaining access to this antifungal drug for many Asian patients, combined with the limited supply of L-AmB (the formulation which is less toxic), have restricted its utilization in low-income countries, where talaromycosis is endemic. Fortunately, recent investigations [13,32,61,109,110,111] have revealed that amphotericin B is not superior to itraconazole for the clinical treatment of talaromycosis, particularly in terms of mortality or fungicidal activity. Although some analyses have demonstrated that itraconazole alone is less effective than amphotericin B [112], it can plainly be seen that the utilization of itraconazole is likely to assist in the management of *T. marneffei* infection where amphotericin B is unavailable. Additionally, voriconazole represents another viable alternative to amphotericin B, as few differences in medication efficacy have been reported [113]. In other words, the actions of voriconazole and amphotericin B are similar as clinical therapeutic options. In these circumstances, azoles (itraconazole and voriconazole) can and are administered with close monitoring, when amphotericin B is unavailable.

Other than azoles, other alternative drugs may also be explored. For example, it is known that clinical antiviral medications which target HIV aspartyl protease (indinavir, saquinavir, ritonavir, lopinavir, etc.) have an inhibitory effect on *Candida albicans* (*C. albicans*) [114]. It is also known that *pop* genes encode aspartyl proteases and have been linked to *T. marneffei* intracellular proliferation after infection [115]. Therefore, one may speculate on the potential suppressive effects mediated by antiretroviral drugs on *T. marneffei* growth in the host. Additionally, it has been demonstrated that galactose exerts growth-inhibitory effects on clinical *T. marneffei* isolates [116]. This finding is significant, as a culture of *T. marneffei* in the presence of galactose may lead to the creation of avirulent strains of *T. marneffei*, which may be important for the development of vaccines against *T. marneffei*, as suggested by Wang et al. [29].

Investigations are ongoing in the quest to discover novel therapeutic candidates against *T. marneffei* [117,118,119,120,121]. For example, one new antifungal drug, named olorofim, is currently being studied in a phase IIb clinical trial [119]. Another agent, named osthole (from traditional Chinese medicine), has demonstrated antifungal activity against *T. marneffei* [120]. Furthermore, it has been shown that a hexane extract (AMA50CH) from a marine-derived actinomycete was capable of neutralizing *T. marneffei* in vitro Finally, purified proteins from two medicinal plants (*Andrographis paniculata* and *Rhinacanthus nasutus*) have been shown to have the capacity to kill *T. marneffei* in vitro [117,121].

#### 3.3.2. Potential Clinical Drug Resistance

Subsequent to clinical treatment, a recurrence or relapse can occur in *T. marneffei* infected individuals [122]. Thus, long-term prophylaxis with oral itraconazole (for example) are routinely required to consolidate treatment and prevent the recurrence of *T. marneffei* infection. Such prophylactic and/or therapeutic approaches might continue from a week to months, and the long-term exposure of fungal isolates to antifungal drugs are potential opportunities for the development of drug resistance. Fortunately, no clinical drug-resistant *T. marneffei* isolates have been identified as yet [29]. Nevertheless, efforts are ongoing to better understand *T. marneffei* and potential drug resistance pathways. For example, Utami et al. [123] have observed that *T. marneffei* multidrug resistance 1 and 3 (PmMDR1 and PmMDR3), which are major facilitator superfamily transporters of *T. marneffei*, are involved in drug resistance to azoles, pyrimidine analogs, and antimalarials. Thus, although no clinical drug-resistant *T. marneffei* isolates have been discovered as yet, researchers have reported that *T. marneffei* does possess the capacity to develop drug resistance, just as other fungi do [124,125].

#### 3.3.3. Coinfection Issues

The treatment of *T. marneffei* infection in HIV-infected individuals is challenging. Due to their immune systems being inherently compromised, it is possible to encounter coinfections with candidiasis (42%) [126], *Pneumocystis* pneumonia (19.9%) [126], tuberculosis (TB) (15.3%) [126], cytomegalovirus (17.3%) [41], salmonellosis (3.5%) [13], and cutaneous herpes (4.6%) [13]. These coinfections associated with *T. marneffei* infection represent a critical challenge, as each of these coinfections are individually potentially lethal. For example, TB, cryptococcosis, and talaromycosis are each known for their deadly consequences, and as such have been considered to be the top three opportunistic illnesses in HIV-infected individuals by some authors [10]. Thus, it is possible that in a coinfection milieu, different pharmacological sensitivities arise, particularly in cases where multiple fungal co-pathogens are identified. For example, it has been reported that HIV-positive patients can present with a coinfection with *T. marneffei* and *C. neoformans* [127]. In this particular instance, Le et al. [127] have noted that the different pharmacological sensitivities of *T. marneffei* and *C. neoformans* to fluconazole and itraconazole have created serious therapeutic issues in clinical settings. The management of such a scenario in resource-limited regions, where these drugs have limited availability or are unavailable, may create an endless therapeutic nightmare for patients.

In addition to HIV-infected individuals, coinfections may be seen in HIV-negative individuals. For example, coinfection with nontuberculous mycobacteria [128], *Mycobacterium tuberculosis* [129], and *C. neoformans* [127,130]. Interestingly, in HIV-negative individuals, coinfection by *T. marneffei* and nontuberculous mycobacteria have become more common of late [9,131,132]. In this context, the diagnosis of *T. marneffei* infection may be much more complex, and management of the case may also expose different pharmacological sensitivities and potential drug–drug interactions.

A list of drugs potentially useful for the prophylactic approach and treatment of *T. marneffei* infection is provided in Table 2.

## 4. Conclusions

The available diagnostic and therapeutic options for talaromycosis represent two major determinants for the effective control of the infection. On the one hand, despite the existence of numerous diagnostic options, the rapid and accurate diagnosis of talaromycosis remains challenging. The most efficient approach to diagnosis relies on the use of mNGS, which is expensive and is not available in poor socioeconomic settings. Due to the unknown duration of the incubation period for *T. marneffei*, it remains challenging to diagnose the disease at its early stage. This issue is of particular concern, especially in clinical practice, where the prognosis of patients is entirely dependent on early diagnosis. On the other hand, several existing antifungal drugs are being used to treat *T. marneffei* infection. However, since the case management requires long-term treatment, more attention should be focused on the development of novel antifungal drugs to avoid the evolution and spread of strains resistant to current therapeutic drugs. Furthermore, sustained efforts should be devoted toward the investigation and discovery of viable animal models of talaromycosis, which would be invaluable in the quest for a more comprehensive appreciation of the underlying immunopathogenic processes which are inherent to *T. marneffei* infection. We sincerely hope that researchers explore innovative approaches to more effectively prevent *T. marneffei* infection in endemic areas, for example, through the development of vaccines against *T. marneffei*, or the analysis and discovery of highly specific biomarkers for *T. marneffei*, which are essential for the rapid and accurate diagnosis of *T. marneffei* infection.

## Figures and Tables

**Figure 1 jof-09-00647-f001:**
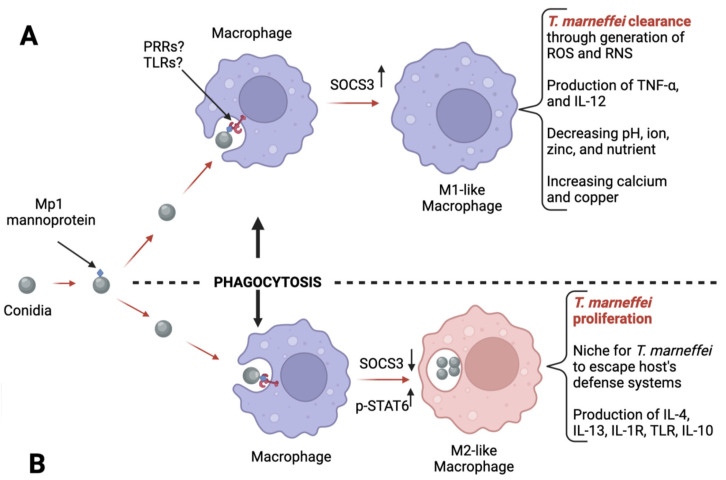
*T. marneffei* interaction with host macrophages. The infectious fungal conidia provoke infection. The Mp1 mannoprotein in the conidium wall is recognized by macrophages via unidentified PRRs or TLRs [74]. Phagocytosis of the conidium is subsequently initiated by the macrophage. In case (**A**), inflammatory cytokines such as IFN-γ may stimulate the formation of M1-like macrophages, which kill *T. marneffei* through a generation of reactive oxygen species (ROS), and reactive nitrogen species (RNS). Furthermore, there is production of pro-inflammatory cytokines such as TNF-α and IL-12 to recruit more M1-like macrophages. In case (**B**), instead of pro-inflammatory cytokines, it has been noted that anti-inflammatory proteins such as IL-4, IL-13, and IL10, and expression of receptors such as IL-1R and TLR are observed. These cytokines promote M2-like macrophage proliferation. Unfortunately, M2-like macrophages do not kill *T. marneffei*; however, they provide a niche for *T. marneffei* to evade the host’s defense systems. This behavior of M2-like macrophages is referred to as the macrophage paradox effect. Once within M2-like macrophage, *T. marneffei* may induce the promotion of more M2-like macrophages via the degradation of M1-like polarization factor SOCS3. Via this inhibitory activity exerted towards SOCS3, *T. marneffei* relieves its inhibitory effect on the M2-like macrophage polarization factor p-STAT6 [75]. A diagnostic approach based on the profiling of polarization factor SOCS3 or p-STAT6 would be interesting to explore.

**Table 1 jof-09-00647-t001:** Current approaches to diagnose *T. marneffei* infection.

Diagnosis Approach	Specimen	Target	References	Advantages	Disadvantages
Culture-based	Blood, skin, bone marrow, or lymph node biopsy	Physical presence of *T. marneffei*	[13,35,61]	High specificity	Too slow, delays therapeutic intervention, and limited sensitivity (disseminated infection)
Microscopy	Blood, skin, bone marrow, or lymph node biopsy	Physical presence of *T. marneffei*	[35,40]	Quick to perform and high specificity	Requires highly trained microscopists (at least two) and has limited sensitivity (when skin is considered)
Antigen/antibody	Blood or urine	Monoclonal antibody (mAb) EB-A1	[42,43]	Quick to perform, high sensitivity, and specificity	Potential cross-reactions (galactomannan also found in *Aspergillus* spp. and *Cryptococcus neoformans*, or elevated levels of β-D-glucan also reported in aspergillosis and candidiasis), and efficacy depends on the specimen used
		mAb-4D1	[46,47]		
		mAb-Mp1p	[36,48]		
PCR-based and mNGS	Blood, skin, bone marrow, lymph nodes, or formalin-fixed and paraffin-embedded (FFPE) samples	5.8S rRNA	[49]	Quick to perform, high sensitivity, and high specificity	Expensive and unavailable in poor socioeconomic settings
		18S rRNA	[50]		
		MP1	[51]		

Ab: antibody; MP1: mannoprotein 1.

**Table 2 jof-09-00647-t002:** Reported therapeutic options for *T. marneffei* infection.

Purpose	Drug Name	Mechanism of Action	*T. marneffei* Reported Resistance	References
Prophylaxis and treatment	Itraconazole	Inhibits cytochrome P(CYP)-450-dependent enzymes, which induces the impairment of ergosterol synthesis	PmMDR1 and PmMDR3 pathways are potential mechanisms	[123,133]
Prophylaxis	Co-trimoxazole	Blocks dihydropteroic acid synthase (DHPS), dihydrofolate synthase (DHFS), and dihydrofolate reductase (DHFR)	PmMDR1 and PmMDR3 pathways are potential mechanisms	
Treatment	Amphotericin B *****	Binds to sterols in the cell membrane and creates a trans membrane channel, which allows leakage of intracellular components	N/A	[98,134]
	Voriconazole	Binds to CYP51 and inhibits the demethylation of lanosterol (and ergosterols in general). Thus, the lack of ergosterol leads to disruption of the cell membrane, which consequently limits *T. marneffei* growth.	PmMDR1 and PmMDR3 pathways are potential mechanisms	[104,135]
Promising alternatives	Antiviral medications	Inhibits aspartyl protease	N/A	[114,115]
	Olorofim	Selectively inhibits fungal dehydrogenase (DHODH), which is a key enzyme involved in the de novo pyrimidine biosynthesis pathway	N/A	[119,136]
	Fosmanogepix	Inhibits fungal GWT1 protein, which is essential for trafficking and anchoring mannoproteins to the cell membrane and outer cell wall. Consequently, cell wall integrity is disrupted and fungal pathogenicity is considerably neutralized. This drug is currently under investigation in a clinical trial (NCT03604705)	N/A	[137]
	Rezafungin	Disrupts the cell wall of fungal species by inhibiting the 1,3-β-D-glucan synthase enzyme complex present in cell walls of fungi	N/A	[138]
	Isavuconazole	Inhibits the biosynthesis of ergosterol through inhibition of lanosterol 14-alpha-demethylase (a cytochrome P-450 dependent enzyme) which mediates the conversion of lanosterol to ergosterol. The lack of ergosterol leads to disruption of the cell membrane	PmMDR1 and PmMDR3 pathways are potential mechanisms	[139]
	Galactose	Inhibits the growth of *T. marneffei* isolates	N/A	[116]
	Osthole	Inhibits hyphal growth through glucose starvation. It also inhibits spore germination and mycelia growth	N/A	[120,140]
	AMA50CH	Binds to the cell membrane and subsequently promotes broken cells with leak-out of the intracellular content.	N/A	[118]
	Proteins from *Andrographis paniculata*	Activate the heterotrimeric G-proteins and lead to apoptotic-like death of the fungal cells	N/A	[121]
	Proteins from *Rhinacanthus nasutus*	Activate the heterotrimeric G-proteins and lead to apoptotic-like death of the fungal cells	N/A	[117,121]

N/A: not reported; *****: First-line antifungal treatment for *T. marneffei* infection.

## Data Availability

Not applicable.

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
