# Peer review of "An Overview of Diagnostic and Management Strategies for Talaromycosis, an Underrated Disease"

_jof, 2023, doi:10.3390/jof9060647_

Round 1

Reviewer 1 Report

The authors provide a review of diagnostics and management strategies for talaromycosis. A few comments for the author's consideration:

1. It would be helpful to readers to mention in the Introduction that the former name for Talaromyces marneffei was Penicillium marneffei and much of the older literature uses that name. You do mention the old disease name of Penicilliosis but don't give the old organism name anywhere.

2. Introduction line 49 - is T. marneffei generally accepted to be spread person-to-person or is it environmentally acquired? The sentence as written is confusing.

3. Introduction line 72 - define the abbreviation PLWH;

4. Diagnosis line 112-115 - the text reads as if dimorphic growth is a diagnostic disadvantage. Can you expand on why in line 129 or perhaps just adjust the text to note that the two forms that grow at the different temperatures?

5. Diagnosis line 129 - I would emphasize for the reader who is new to this topic that a central septation and division by fission is in contrast to how most other clinically significant yeast reproduce which is by budding.

6. Diagnosis line 172-200 - suggest mentioning that both PCR and mNGS are technically complex to perform.

7. Diagnosis - there is no discussion of antifungal susceptibility testing; a brief description of the recommended method(s) and any thing known about breakpoints or epidemiological cutoff values would add to the comprehensiveness of the review.

8. Treatment, currative approaches - please comment on what is known about the activity of some of the newer antifungal agents such as olorofim, fosmanogepix, rezafungin, and isavuconazole. Olorofim is listed in Table 2 but the other agents are not mentioned.

9. Coinfection issues line 455 - can the authors provide references for the statement that coinfection with NTM is increasing?

Author Response

Reviewer 1

The authors provide a review of diagnostics and management strategies for talaromycosis. A few comments for the author's consideration:

  1. It would be helpful to readers to mention in the Introduction that the former name for Talaromyces marneffei was Penicillium marneffei and much of the older literature uses that name. You do mention the old disease name of Penicilliosis but don't give the old organism name anywhere.

Answer: Thank you. The former name for Talaromyces marneffei, which was Penicillium marneffei, has now been included in the text (Pages 1-2, lines 29-30).

  1. Introduction line 49 - is T. marneffei generally accepted to be spread person-to-person or is it environmentally acquired? The sentence as written is confusing.

Answer: Thank you. The irksome sentence has now been altered (Page 2, lines 51-53).

  1. Introduction line 72 - define the abbreviation PLWH;

Answer: Thank you. The definition of the abbreviation ‘PLWH’ has now been provided in the text (Page 3, line 80)

  1. Diagnosis line 112-115 - the text reads as if dimorphic growth is a diagnostic disadvantage. Can you expand on why in line 129 or perhaps just adjust the text to note that the two forms that grow at the different temperatures?

Answer: Thank you. The text has been adjusted, as suggested (Page 4, lines 123-124).

  1. Diagnosis line 129 - I would emphasize for the reader who is new to this topic that a central septation and division by fission is in contrast to how most other clinically significant yeast reproduce which is by budding.

Answer: Thank you. The text has been further adjusted to reflect this (Page 4, lines 142-143).

  1. Diagnosis line 172-200 - suggest mentioning that both PCR and mNGS are technically complex to perform.

Answer: Thank you. The text has been further adjusted, as suggested (Page 6, lines 197-198).

  1. Diagnosis - there is no discussion of antifungal susceptibility testing; a brief description of the recommended method(s) and any thing known about breakpoints or epidemiological cutoff values would add to the comprehensiveness of the review.

Answer: Thank you. A discussion on antifungal susceptibility testing has now been included in the text (Page 11, lines 324-340).

  1. Treatment, currative approaches - please comment on what is known about the activity of some of the newer antifungal agents such as olorofim, fosmanogepix, rezafungin, and isavuconazole. Olorofim is listed in Table 2 but the other agents are not mentioned.

Answer: Thank you. Table 2 has now been modified, and newer antifungal agents such as fosmanogepix, rezafungin, and isavuconazole are now included.

  1. Coinfection issues line 455 - can the authors provide references for the statement that coinfection with NTM is increasing?

Answer: Thank you. The required references have now been added, as suggested (Page 16, line 519).

Reviewer 2 Report

This is a  comprehensive review. 

I have some minor comments

It would be useful in the introduction to comment on a) the current status of HIV in the endemic areas b) comment on other predisposing factors such as the increasing use of biologic medicines

In discussing diagnosis please indicate the potential cross reactive or co-endemic organisms that may be confused

What is the current advice on maintaining treatment in patients with HIV ie when to stop ?

Some minor editorial changes are required

Author Response

Reviewer 2

This is a  comprehensive review. 

I have some minor comments

It would be useful in the introduction to comment on

  1. the current status of HIV in the endemic areas b) comment on other predisposing factors such as the increasing use of biologic medicines

Answer: Thank you. The current status of HIV in endemic areas has now been included. Furthermore, the other predisposing factors, such as increasing use of biologic medicines, have now been added to the text (Page 2, lines 56-61).

In discussing diagnosis please indicate the potential cross reactive or co-endemic organisms that may be confused

Answer: Thank you. The requested information has been included in Table 1. 

What is the current advice on maintaining treatment in patients with HIV ie when to stop ?

Answer: Thank you. During HIV, treatment for talaromycosis should be followed by prophylactic treatment to prevent relapse. Currently, as recommended by Tun et al., [1] secondary prophylaxis can be safely stopped in patients with talaromycosis after immune reconstitution, with a sustained increase in CD4+ T-cell counts to ≥100 cells/µL after ART initiation. This information has now been added to the text of the manuscript (Page 11, lines 351-355).

References

[1] N Tun, A Mclean, X Deed, M Hlaing, Y Aung, E Wilkins, E Ashley, and F Smithuis. Is stopping secondary prophylaxis safe in HIV-positive talaromycosis patients? Experience from Myanmar. HIV Med. 2020 Nov; 21(10): 671–673.
